# Facile Aqueous-Phase Synthesis of Bimetallic (AgPt, AgPd, and CuPt) and Trimetallic (AgCuPt) Nanoparticles

**DOI:** 10.3390/ma13020254

**Published:** 2020-01-07

**Authors:** Zengmin Tang, Euiyoung Jung, Yejin Jang, Suk Ho Bhang, Jinheung Kim, Woo-Sik Kim, Taekyung Yu

**Affiliations:** 1Department of Chemical Engineering, College of Engineering, Kyung Hee University, Yongin 17104, Korea; wufengtzm@khu.ac.kr (Z.T.); jey9207@khu.ac.kr (E.J.); 2Department of Chemistry and Nano Science, Ewha Womans University, Seoul 03760, Korea; loloranze@ewhain.net; 3School of Chemical Engineering, Sungkyunkwan University, Suwon 16419, Korea; sukhobhang@skku.edu

**Keywords:** multi-metallic, noble metals, aqueous-phase, nanoparticles, co-reduction

## Abstract

Multi-metallic nanoparticles continue to attract attention, due to their great potential in various applications. In this paper, we report a facile aqueous-phase synthesis for multi-metallic nanoparticles, including AgPt, AgPd, CuPt, and AgCuPt, by a co-reduction method within a short reaction time of 10 min. The atomic ratio of bimetallic nanoparticles was easily controlled by varying the ratio of each precursor. In addition, we found that AgCuPt trimetallic nanoparticles had a core-shell structure with an Ag core and CuPt shell.

## 1. Introduction

Currently, multi-metallic nanoparticles, as opposed to mono-metallic nanoparticles, have attracted more attention, with a steady flow of research in various applications, such as sensors, energy conversion and storage, biomedicine, and catalysts [1,2,3,4,5]. Merging two or more metals into one nanoparticle is an effective way to generate two well-known effects. Firstly, the electronic structure of the multi-metallic particles can be modulated repeatedly, due to different binding forces on electrons between various metal atoms, thus enhancing their catalytic performance in catalysis [6,7]. Secondly, the synergistic effect is probably brought about by the appropriate combination of metals and the atomic ratio of various metals, eventually attaining high catalytic efficiency, including high catalytic activity and high selectivity [8,9,10,11]. For example, Kim and co-workers showed that AuCu alloy nanoparticles exhibited better catalytic activity than Au or Cu in the electrochemical reduction of carbon dioxide, because the alloy nanoparticles changed surface composition, thus affecting the d-band and atomic arrangement at the active sites [10]. Hu and co-workers could also obtain higher electrochemical performance for formic acid oxidation by using PdNiCu trimetallic alloy nanoparticles as electrocatalysts [11]. Due to the different electronic structure compared with Pd and PdCu, PdNiCu alloys revealed low poisoning and better stability than Pd and PdCu nanoparticles.

Many researchers have attempted to synthesize multi-metallic nanoparticles, even though their preparation is more complicated compared to mono-metallic nanoparticles [12,13]. In the wet chemical approach, typically three synthetic routes can be used to obtain multi-metallic nanoparticles: seed-mediated growth, galvanic replacement, and co-reduction. Seed-mediated growth, which happens with a large difference in reaction rate between two metal precursors, is used to synthesize multi-metallic nanoparticles with core-shell structures (Pd@Pt) and bimodal nanoparticles (Janus structure) [14,15,16]. Galvanic replacement, which involves a corrosion process driven by the different electrochemical potentials between two metallic species, leads to the formation of hollow and frame nanostructures [17,18,19]. The co-reduction method, which simultaneously reduces two metal salts, is a straightforward method to obtain multi-metallic nanoparticles [1,20]. Among them, the chemical co-reduction strategy attracts attention, due to ease in the control of size, shape, and the ratio of multi-metallic species in the nanoparticles. Chen et al. applied the co-reduction route to synthesize PtAu nanocubes with high shape selectivity in an aqueous polyallylamine hydrochloride (PAH) solution, and found that PAH was a good shape selective agent to control the morphology, composition, and size distribution of the PtAu nanoparticles. However, a high reaction temperature (120 °C) and long reaction time (6 h) was required, due to the use of a weak reducing agent (HCHO) [21]. In the case of tri-metallic nanoparticles, mesoporous PtPdNi alloy nanoparticles could be also fabricated by co-reduction of three metal precursors in a water bath at 40 °C for 4 h under constant sonication, in the presence of Pluronic F127 (PEO_100_PPO_65_PEO_100_) as the structure-directing agent [22]. However, sonication often suffers from high energy consumption and difficulty for mass production. In addition, most of the previous synthetic methods for alloyed nanomaterials have required a long synthesis time to obtain the product.

Herein, we report a facile aqueous-phase synthetic route to various multi-metallic nanoparticles, including AgPt, AgPd, CuPt, and AgCuPt. The synthesis was performed by co-reduction of the metal precursor with L-ascorbic acid in the presence of branched polyethyleneimine (BPEI) as a stabilizer for a short reaction time (10 min). By detailed characterization of the synthesized nanoparticles, we found that the AgPt, AgPd, and CuPt nanoparticles had an alloy form, while AgCuPt had a core-shell structure with an Ag core and CuPt shell. The atomic ratio of metals in the bimetallic nanoparticles was easily controlled by modulating the ratio of each precursor.

## 2. Materials and Methods

The materials used—silver nitrate (AgNO_3_), cupric nitrate hydrate (Cu(NO_3_)_2_·3H_2_O), potassium tetrachloroplatinate (K_2_PtCl_4_, 99.99%), sodium tetrachloropalladate (Na_2_PdCl_4_, 98%), branched polyethyleneimine (BPEI; molecular weight (MW) = 750,000; 50 wt. % solution in water), and L-ascorbic acid (C_6_H_8_O_6_ ≥ 99%)—were purchased from Sigma-Aldrich. All chemicals were used as received without further purification.

To produce AgPt nanoparticles, 20 mg of BPEI were dissolved in 5 mL of deionized (DI)-water in a 20-mL vial under vigorous magnetic stirring. The AgNO_3_ solution (20 μL, 1 M) and a certain volume of 10 mM K_2_PtCl_4_ solution (50 μL, 200 μL, 300 μL, 400 μL, 1000 μL, and 2000 μL) were simultaneously added into the BPEI solution. Then, the above mixture in the vial was heated at 80 °C in oil bath and stirred for complete mixing. A 3 mL aliquot of ascorbic acid (0.2 M) solution was injected using a pipette. After heating at the same temperature for 10 min, the resulting mixture was cooled down to room temperature, transferred into a 50 mL tube, and centrifuged at 8000 rpm for 10 min to get the products precipitate. The products precipitate was purified by repeat centrifugation and washing three times with water to remove excess stabilizer BPEI. The purified products were labeled as AgPt-40, AgPt-10, AgPt-6.7, AgPt-5, AgPt-2, and AgPt-1, respectively, depending on the molar ratio of Ag/Pt precursors. For transmission electron microscopy (TEM) analysis, the purified precipitate was re-dispersed in 5 mL of water, and 4 μL of purified colloidal products solution was placed on nickel TEM grid and dried in the drying oven for 1 h to prepare the sample for TEM measurement. For inductively coupled plasma spectrometer (ICP) analysis, 1 mL of purified colloidal products solution was completely dissolved into nitrohydrochloric acid (HCl:HNO_3_ = 3:1) and diluted to 10 mL using water.

Synthesis of AgPd bimetallic nanoparticles: In a 20 mL of vial, 2 mg of BPEI was dissolved in 5 mL of water to compose the BPEI solution. Then 20 μL of AgNO_3_ solution (1 M) and a certain volume of 65 mM Na_2_PdCl_4_ (200 μL for AgPd-1.3, 300 μL for AgPd-1, 400 μL for AgPd-0.8, and 500 μL for AgPd-0.66) were added into the BPEI solution. After the reacting mixture solution was heated to 80 °C, 3 mL of L-ascorbic acid (0.2 M) was then added. The subsequent experiment is the same as the manufacturing method for AgPt bimetallic nanoparticles.

For the synthesis of CuPt bimetallic nanoparticles, 10 mg of BPEI was added in 5 mL of water in a 20-mL vial for the BPEI solution. To form the bimetallic CuPt nanoparticles, 1000 μL of K_2_PtCl_4_ (10 mM) solution and a certain volume of 1 M Cu(NO_3_)_2_ solution (200 μL for PtCu-20, 100 μL for PtCu-10, and 40 μL forPtCu-4) were added into the BPEI solution. The subsequent experiment is the same as the manufacturing method for AgPt bimetallic nanoparticles.

For the synthesis of AgCuPt trimetallic nanoparticles, 20 mg of BPEI was dissolved in 5 mL of water for the BPEI solution. A 20 μL aliquot of AgNO_3_ solution (1 M), 1000 μL of K_2_PtCl_4_ (10 mM) solution, and 100 μL of Cu(NO_3_)_2_ (1 M) solution (the atomic ratio of Cu:Ag:Pt was 10:2:1) were added into the BPEI solution. The subsequent experiment is the same as the manufacturing method for AgPt bimetallic nanoparticles.

X-ray diffraction (XRD) patterns were measured by using a Rigaku D-MAX/A diffractometer (Bruker, Germany) at 35 kV and 35 mA. UV-Vis measurements were conducted on a Shimadzu 2550 spectrophotometer (Shimadzu, Kyoto, Japan). Transmission electron microscopy (TEM) and energy-dispersive X-ray spectroscopy (EDS) mapping images were captured by using a JEM-2100F transmission electron microscope (JEOL, Tokyo, Japan) at 200 kV. The nanoparticle size was measured from each particle’s TEM image. We measured the size of 200 particles per TEM image and expressed it as a histogram. Metal inductively coupled plasma spectrometer (ICP) analyses were performed by using a Direct Reading Echelle ICP (Leeman, Hudson, NH, USA). Zeta-potential was performed by using Nano-ZS (Malvern, UK).

## 3. Results

AgPt nanoparticles were synthesized by co-reducing AgNO_3_ and K_2_PtCl_4_ with L-ascorbic acid in the presence of BPEI as a stabilizer and in an aqueous phase at 80 °C for a short reaction time of 10 min. For atomic ratio control between Ag and Pt, we modulated the concentration of the Pt precursor while holding other experimental conditions constant. In this synthesis, we labeled the samples as AgPt-40, AgPt-10, AgPt-6.7, AgPt-5 AgPt-2, and AgPt-1. Typical TEM images of the AgPt nanoparticles demonstrate that all of the synthesized nanoparticles had spherical shapes, with sizes of 30.09 ± 6.30 nm for AgPt-40, 47.06 ± 8.04 nm for AgPt-10, 49.83 ± 9.15 nm for AgPt-6.7, 54.88 ± 10.67 nm for AgPt-5, 30.04 ± 7.71 nm for AgPt-2, and 31.21 ± 9.62 nm for AgPt-1 (Figure 1 and Appendix A). The atomic percentage of Pt in the nanoparticles characterized by an inductively coupled plasma spectrometer (ICP) analysis were 1.24 at. % for AgPt-40, 5.58 at. % for AgPt-10, 10.12 nm at. % AgPt-6.7, 12.85 nm at. % AgPt-5, 21.24 at. % for AgPt-2, and 30.7 at. % for AgPt-1, indicating that the amount of Pt in the nanoparticles increased as the amount of Pt precursor used increased (Appendix A and Appendix A). In the synthesis of AgPt nanoparticles, the yield of Ag almost was 100%, and yield of Pt was a little low, with a range of 23.8% to 41.0%. Interestingly, the size of the nanoparticles tended to increase as the proportion of Pt in nanoparticle increased to 12.85%, and then decreased again. We believe that this phenomenon is due to the combined effect of the different reducing rate between the Ag and Pt precursor and the presence of the galvanic replacement reaction [23,24]. Energy dispersive X-ray spectroscopy (EDS) mapping images of AgPt nanoparticles show that both Ag and Pt were well dispersed in the nanoparticles, indicating the formation of the alloy structure (Appendix A). A lattice distance of the AgPt-6.7 nanoparticle shown in Appendix A was 2.30 Å and assigned to (111) plane, which was between the (111) plane of Ag (2.35 Å) and (111) plane of Pt (2.26 Å), due to effect of combination between Ag and Pt. As a part of the efforts to obtain a better understanding of the formation of alloy structures, we measured X-ray diffraction (XRD) and UV-Vis analyses using the AgPt nanoparticles. The XRD results show that AgPt-40 nanoparticles were similar to face-centered cubic (*fcc*) Ag (Fm3m, *a* = 4.086 Å; Joint Committee on Power Diffraction Standard (JCPDS) #04-0783), and XRD peaks gradually right-shifted toward Pt by increasing Pt in the nanoparticles (Appendix A). Furthermore, UV-vis spectra reveal that the localized surface plasmon resonance (LSPR) peak position of AgPt-40 nanoparticles was 481 nm (Appendix A). As the amount of Pt increased, the LSPR peak positions were red-shifted and eventually disappeared, supporting the formation of AgPt alloy in the present synthesis.

We also investigated the synthesis of AgPd nanoparticles by using BPEI, L-ascorbic acid, AgNO_3_, and Na_2_PdCl_4_ as similar synthetic conditions for AgPt nanoparticles. Figure 2 shows typical TEM images of the AgPd nanoparticles, indicating that AgPd-1, AgPd-0.8, and AgPd-0.66 had uniform sizes of 14.80 ± 2.93 nm, 19.22 ± 2.98 nm, and 21.91 ± 3.64 nm, respectively (Appendix A). In the case of AgPd-1.3, it was observed that the size was slightly uneven and mixed with about 10% rod-shaped structures (Figure 2a). From EDS mapping images (Appendix A), we observed that Ag and Pd atoms were well distributed in the nanoparticles, thus confirming the formation of the alloy structure. Due to the formation of the alloy, the lattice distance of the AgPd-1 alloy was 2.29 Å, which did not match with the (111) plane of Ag (2.35 Å) and the (111) plane of Pd (2.24 Å) (Appendix A). The percentage of Pd in the AgPd-1.3, AgPd-1, AgPd-0.8, and AgPd-0.66 nanoparticles was 2.1%, 17.5%, 21.7%, and 34.4%, respectively, and corresponded to the ICP results (Appendix A). The yield of Pd in the synthesis of AgPd also increased from 2.9% to 31.8%, and the yield of Ag was closed to 100% (Appendix A).

The XRD analyses show that the crystal structure of the AgPd-1.3 nanoparticles was similar to *fcc* Ag (Fm3m; *a* = 4.086 Å; JCPDS #04-0783), due to a larger proportion of Ag than Pd in the nanoparticles. By increasing the amount of Pd, AgPd-1, AgPd-0.8, and AgPd-0.66 exhibited a peak-shift compared to pure Ag, revealing the formation of the AgPd alloy (Appendix A). Similar to the AgPt nanoparticles, the disappearance of an LSPR peak at 452 nm as the ratio of Pd increased indicated the formation of the alloy structure, not bimodal particles (Appendix A).

Typically, because Cu can be easily oxidized by water and oxygen in the air, and its reduction potential is very different from the reduction potential of Pt or Ag, it is difficult to synthesize a Cu-based alloy compared with Ag-based alloy nanoparticles in an aqueous phase [25,26]. In a previous report on the aqueous-phase synthesis of Cu nanocrystals with long-term stability in air, it was demonstrated that BPEI formed a Cu–BPEI complex, thus leading to the formation of stable Cu nanocrystals in an aqueous phase [27,28]. In the present synthesis, we attempted to synthesize CuPt nanoparticles in an aqueous solution using Cu(NO_3_)_2_ and K_2_PtCl_4_ in the presence of BPEI. Figure 3 shows typical TEM images of CuPt-20, CuPt-10, and CuPt-4 nanoparticles, indicating that they had spherical shapes and uniform sizes of 12.62 ± 1.95 nm, 14.13 ± 1.81 nm, and 23.80 ± 3.47 nm, respectively (Appendix A). We investigated the XRD to check the structure of CuPt nanoparticles, confirming that *fcc* Cu (Fm3m; *a* = 3.615 Å; JCPDS #04-0836) is similar to CuPt-20 nanoparticles (Figure 3d). As the amount of Pt increased, the XRD peaks shifted, demonstrating the formation of the CuPt alloy. Additionally, there was no peak like Cu_2_O in the nanoparticles in the XRD patterns. Similar to AgPt and AgPt, the lattice distance in CuPt-40 was 2.16 Å, which was between the (111) plane of Pt (2.24 Å) and the (111) plane of Cu (2.08 Å), due to the formation of the alloy (Appendix A). In addition, EDS mapping images of all of the CuPt nanoparticles show that Cu and Pt atoms were uniformly dispersed in the nanoparticles, and the atomic ratios of Cu/Pt in the nanoparticles were 91.8:8.2 for CuPt-20, 86.1:13.9 for CuPt-10, and 77.9:22.1 for CuPt-4, respectively (Appendix A). Appendix A shows that the yield of Pt in the synthesis of CuPt nanoparticle is above 80%. The yield of Cu is 34.0%, 52.1%, and 77.5% in CuPt-20, CuPt-1-, and CuPt-4 respectively (Appendix A).

Advancing the synthesis of bimetallic nanoparticles, we attempted to make AgCuPt trimetallic nanoparticles by using the present synthetic method. Synthesized AgCuPt nanoparticles had a spherical shape with an average size of 32.89 ± 4.35 nm (Figure 4a,b). The AgCuPt nanoparticles were composed of 67.6 at. % Cu, 22.2 at. % Ag, and 10.2 at. % Pt, respectively (Appendix A). EDS mapping images reveals that AgCuPt nanoparticles had a core-shell structure with an Ag core and CuPt bimetallic shell, unlike bimetallic nanoparticles (Figure 4c). We thought that Cu and Pt would be reduced after the reduction of Ag, due to the difference of the reduction potentials between Ag, Cu, and Pt precursors (standard reduction potentials were Ag^+^/Ag (0.80 V), Cu^2+^/Cu (0.34 V), and PtCl_4_^2-^/Pt (0.73 V), respectively). XRD analysis also shows the presence of two major crystal structures in the AgCuPt nanoparticles: *fcc* Ag (Fm3m; *a* = 4.086 Å; JCPDS #04-0783) and *fcc* Cu (Fm3m; *a* = 3.615 Å; JCPDS #04-0836) (Figure 4d). The XRD peaks of Cu were slightly left-shifted, demonstrating that it was a CuPt alloy structure. Similar to CuPt-10, due to alloy formation, the lattice distance of shell CuPt was 2.18 Å (Appendix A). The ICP showed that the yield of Cu was 58.7%, Ag was 96.4%, and Pt was 88.7%, respectively.

The stability of the nanoparticles in the solution is also an important factor for the application of nanoparticles. To know the dispersibility of the synthesized nanoparticles, we measured surface potential values of the nanoparticles using zeta potential analysis (Appendix A). Due to the large number of amine groups in BPEI, all of the nanoparticles exhibited positive zeta potential values around 20–30 mV, indicating good dispersibility in water [29].

## 4. Conclusions

In this study, we successfully synthesized bi-metallic (AgPt, AgPd, and CuPt) and tri-metallic (AgCuPt) nanoparticles in an aqueous solution. Notably, the entire procedure was conducted through a facile co-reduction method within a short reaction time (10 min). We could easily control the composition of the nanoparticles by changing the ratio of the two components. We believe that this easy and simple technology to synthesize various multi-metallic noble metal nanoparticles in a short time can greatly contribute to the mass production of nanocatalysts. For their application, the synthesized AgPd and AgPt can be used to enhance catalysts in H_2_O_2_ generation, due to their modified electronic structure compared with individual Pd and Pt. In the case of Cu-based nanoparticles, CuPt and AgCuPt can have high catalytic activity in electrochemical CO_2_ reduction. Therefore, we believe that our new synthetic route, with a lower cost and environmentally-friendly conditions, can be used to pave the way for diverse multi-metallic nanoparticles with enhanced catalytic performance.

## Figures and Tables

**Figure 1 materials-13-00254-f001:**
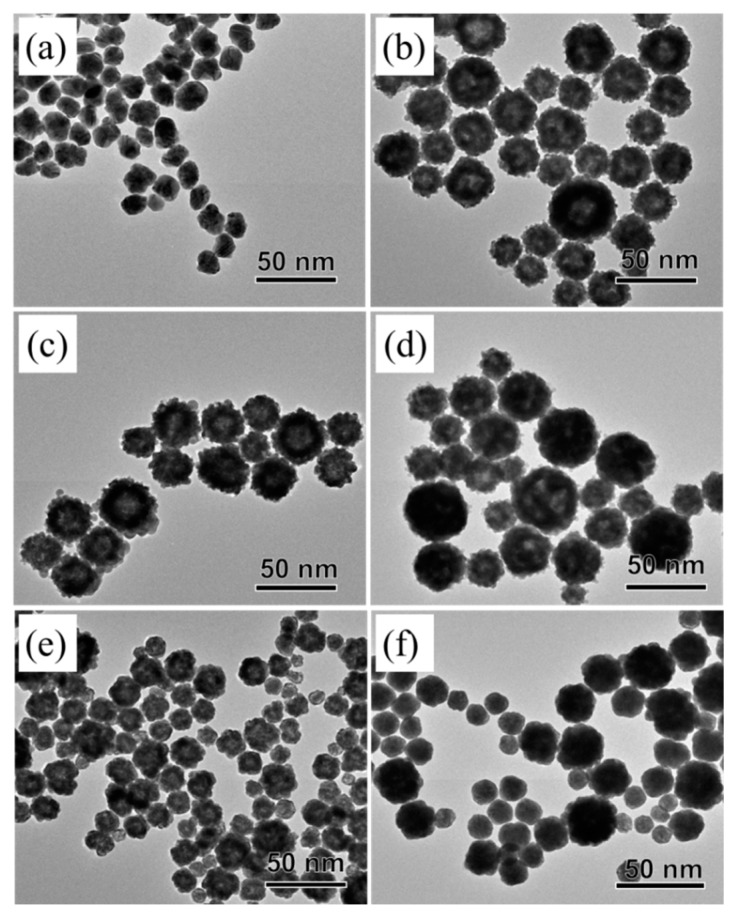
Transmission electron microscopy (TEM) images of AgPt bimetallic nanoparticles at different molar ratios of Ag/Pt in precursor (**a**) AgPt-40, (**b**) AgPt-10, (**c**) AgPt-6.7, (**d**) AgPt-5, (**e**) AgPt-2, and (**f**) AgPt-1.

**Figure 2 materials-13-00254-f002:**
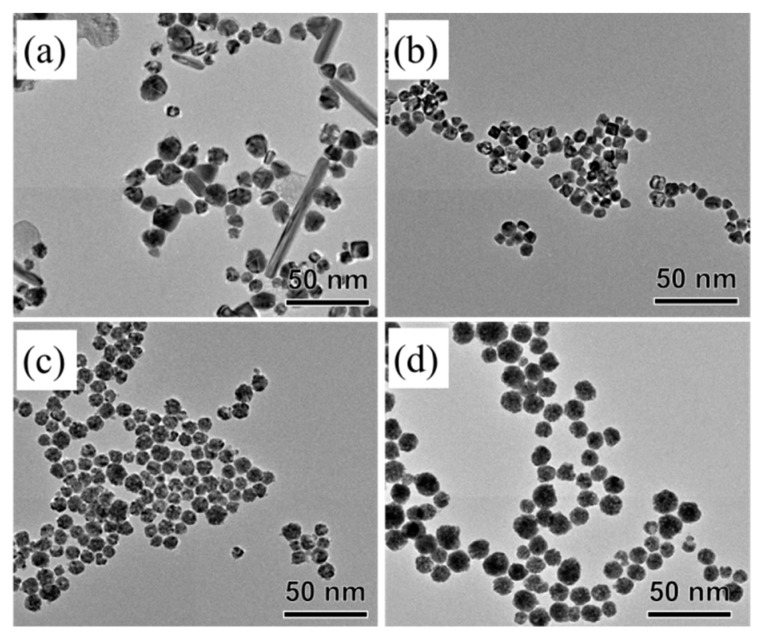
TEM images of as-prepared AgPd nanoparticles: (**a**) AgPd-1.3, (**b**) AgPd-1, (**c**) AgPd-0.8, and (**d**) AgPd-0.66.

**Figure 3 materials-13-00254-f003:**
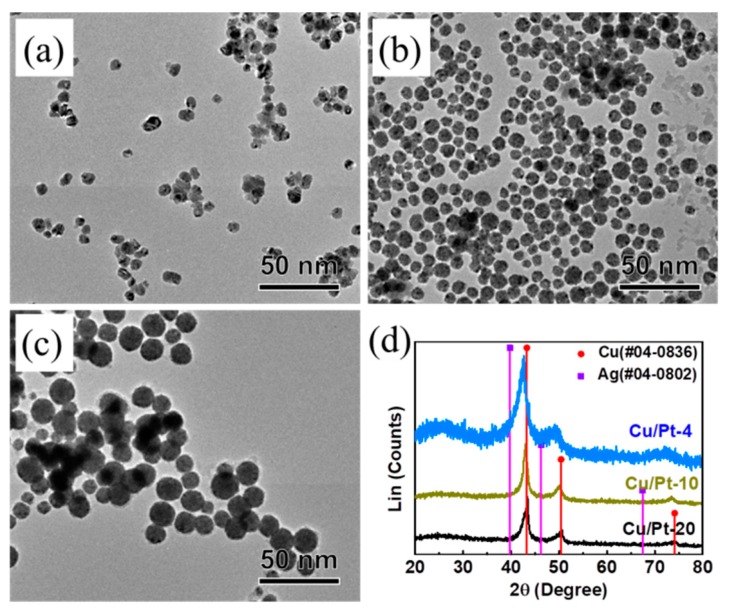
TEM images of as-prepared CuPt nanoparticles: (**a**) CuPt-20, (**b**) CuPt-10, (**c**) CuPt-4, and (**d**) the X-ray diffraction (XRD) pattern of CuPt nanoparticles.

**Figure 4 materials-13-00254-f004:**
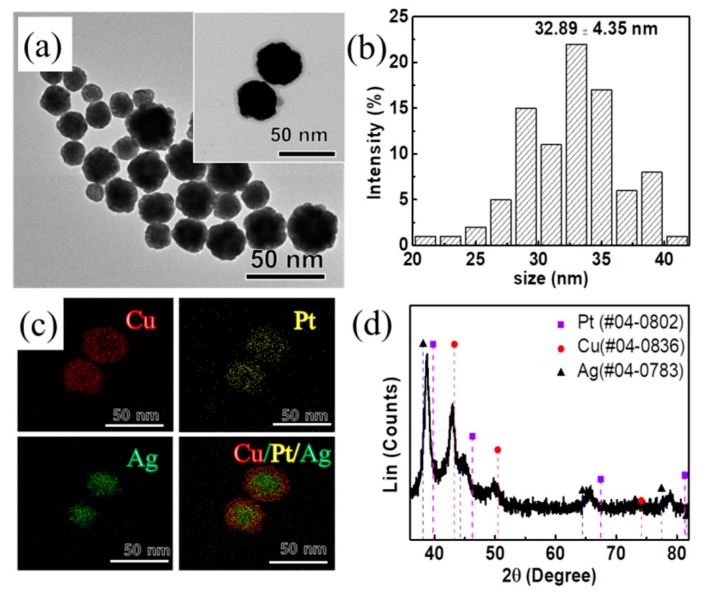
(**a**) TEM image, (**b**) size distribution, (**c**) energy-dispersive X-ray spectroscopy (EDS) mapping images, and (**d**) XRD patterns of metallic elements distribution in as-prepared CuAgPt nanoparticles (red refers to Cu, yellow refers to Pt, and green refers to Ag).

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
