# Peer review of "Facile Aqueous-Phase Synthesis of Bimetallic (AgPt, AgPd, and CuPt) and Trimetallic (AgCuPt) Nanoparticles"

_materials, 2020, doi:10.3390/ma13020254_

Round 1

Reviewer 1 Report

Title: Facile Aqueous-Phase Synthesis of Bimetallic (AgPt, AgPd, and CuPt) and Trimetallic (AgCuPt) Nanoparticles

Authors: Zengmin Tang, Euiyoung Jung, Yejin Jang, Suk Ho Bhang, Jinheung Kim, Woo-Sik Kim, Taekyung Yu

The presented manuscript reports an aqueous-phase synthesis of multi-metallic nanoparticles such as AgPt, AgPd, CuPt, and AgCuPt, using the co-reduction method. The procedure seems to require short reaction times (10 min) and the atom ratio can be tuned by a precursor ratio. Authors provide a series of TEM, SAXS, and EDX data supporting the claims that are being made.

The topic is certainly very interesting, and the aqueous approach offers certain benefits over solvothermal synthesis. I recommend the manuscript for publication, but there are a few points that need to be addressed.

Is only the successful synthesis reported? In other words, can the method be generalized to metals such as Ni and Au? If authors tried other metals or other ratios (even if negative results were obtained), it is highly recommended to include them. The size distribution reported in the text must have a standard deviation. Namely, in a format of ab ± cd nm. It is done so in Fig S1 and some others but must be so everywhere including in the manuscript text as well as in Figure S8. How are the sizes measured? For example, when size is determined, are TEM images used? If yes, authors must report how many particles were measured to determine the average size and such data must be included. If more than one method is used, that needs to be reported as well. Figure S2 and similar figures in the ESI. It is recommended the values on Y-axis be easily comparable to the ones on X-axis. For example, it would be great if atomic ratio values (in NPs) were added to the percentage values. Perhaps in the brackets. Thus, the reader can easily follow what metal ratio was in st. materials vs ended up in the final NPs without doing additional math. Almost all figures in the text need to be adjusted. Namely, the image sizes and text sizes are inconsistent. Also, in some cases, they are unreadably small. Especially the XRD traces and texts on them. The references need polishing. For example, reference 1 and 13 seems to be the same publication. I would suggest the authors elaborate on the application of these NPs. Perhaps in the summary section, they could suggest what the NPs of this size and composition range can be readily used for.

Author Response

Dear Dr. Nevena Chen

Editor of Materials

Dear Dr. Nevena Chen

Many thanks for your e-mail dated Dec. 31th, 2019. We have carefully considered the reviewer’s comments and revised our manuscript accordingly. The changes are highlighted with yellow pen in the revised manuscript. Here we would like to address his/her comments:

Review #1

The presented manuscript reports an aqueous-phase synthesis of multi-metallic nanoparticles such as AgPt, AgPd, CuPt, and AgCuPt, using the co-reduction method. The procedure seems to require short reaction times (10 min) and the atom ratio can be tuned by a precursor ratio. Authors provide a series of TEM, SAXS, and EDX data supporting the claims that are being made. The topic is certainly very interesting, and the aqueous approach offers certain benefits over solvothermal synthesis. I recommend the manuscript for publication, but there are a few points that need to be addressed.

Comment 1. Is only the successful synthesis reported? In other words, can the method be generalized to metals such as Ni and Au? If authors tried other metals or other ratios (even if negative results were obtained), it is highly recommended to include them.

Respose 1. We would like to thank this reviewer’s kind comment. Nonprecious transition metals such as metallic Ni nanoparticle is very unstable in air and water condition, and easily converted to nickel oxide in the water phase. Therefore, synthesis of small nanoparticle including Ni requires special conditions, for example oil phase and N2 atmosphere. (please refer Chen, C. et al., Science 2014, 343, 1339-1343. Li, Q. et al., Nano Energy 2016, 29, 178-197.). Our method is not suitable for synthesis of Ni-rich nanoparticles.

Comment 2. The size distribution reported in the text must have a standard deviation. Namely, in a format of ab ± cd nm. It is done so in Fig S1 and some others but must be so everywhere including in the manuscript text as well as in Figure S8.

Resposne 2. We would like to thank this reviewer’s kind comment. We have revised them in the revised manuscript (Please refer lines 123, 124, 153, 185 and 205).

Comment 3. How are the sizes measured? For example, when size is determined, are TEM images used? If yes, authors must report how many particles were measured to determine the average size and such data must be included. If more than one method is used, that needs to be reported as well.

Response 3. We would like to thank this reviewer’s kind comment. We used TEM images of each sample to measure their size. 200 individual nanoparticles diameters were measured for each sample. The results are expressed by histogram size distribution, average value of the distribution, and the uncertainty as the standard error of the mean (Please refer lines 112 and 113).

Comment 4.Figure S2 and similar figures in the ESI. It is recommended the values on Y-axis be easily comparable to the ones on X-axis. For example, it would be great if atomic ratio values (in NPs) were added to the percentage values. Perhaps in the brackets. Thus, the reader can easily follow what metal ratio was in st. materials vs ended up in the final NPs without doing additional math.

Response 4. We would like to thank this reviewer’s kind comment. The atomic ratio values in nanoparticles (in the brackets) have been added to the percentage values as shown in Figure X1 a and b. (Please refer Figure S2, Figure S9a).

Figure X1. (a)Pt percent in nanoparticle obtained at different atomic ratio of Ag to Pt in precursor. (b) Pd percent in nanoparticle obtained at different atomic ratio of Ag to Pd in precursor. (the atomic ratio of Ag to Pt, and Ag to pd in nanoparticles is shon in the brackets)

Comment 5. Almost all figures in the text need to be adjusted. Namely, the image sizes and text sizes are inconsistent. Also, in some cases, they are unreadably small. Especially the XRD traces and texts on them.

Response 5. We would like to thank this reviewer’s kind comment. In this revised manuscript, the size of all images, text in the images, and XRD trace in XRD pattern have been adjusted to let them easy to read (Please refer Figure 1-4, and Figure S1, S2, S5, S6, S9, S10, and S12)

Comment 6. The references need polishing. For example, reference 1 and 13 seems to be the same publication.

Response 6. We would like to thank this reviewer’s kind comment. In this revised manuscript, repeated reference 13 has been deleted and order of all references has been rearranged.

Comment 7. I would suggest the authors elaborate on the application of these NPs. Perhaps in the summary section, they could suggest what the NPs of this size and composition range can be readily used for.

Response 7. We would like to thank this reviewer’s kind comment. In this revised manuscript, we have commented the application of synthesized nanoparticles. The synthesized AgPd and AgPt can be used to enhanced catalysts in H2O2 generation due to their modified electronic structure compared with individual Pd and Pt. In the case of Cu-based nanoparticles, CuPt and AgCuPt can have high catalytic activity in electrochemical CO2 reduction. (Please refer line 229-231).

Thank you very much for many appropriate and valuable comments. I am sure that these comments improved significantly the quality of the manuscript.

Sincerely yours,

Taekyung Yu

Reviewer 2 Report

The paper from Zengmin Tang et al., titled “Facile Aqueous-Phase Synthesis of Bimetallic (AgPt, AgPd, and CuPt) and Trimetallic (AgCuPt) Nanoparticles, presented a water-based co-reduction methodology to afford bi and tri-metallic nanoparticles. The authors present an alternative route respect to the knowm protocols which exploit the use of L-ascorbic acid and BPEI to synthesize metallic nanoparticles in short reaction time and at not high temperature. The topic is particularly hot, considerig the numbers of different application in which the nanoaprticles can be employed. Therefore, I suggest the pubblication of the presente paper in Materials after some revisions:

In the Materials and Methods part:

1) How the nanoparticles are purified? please comment on it

2) The size is measured by TEM? please describe in details. How many nanoparticles are considered?

3) Describe the method used to prepare the sample for ICP

In the Results part:

1)the size of the nanoparticles must be reported with their standard deviation, for any kind of nanoparticles;

2) I suggest to introduce at least one TEM image for kind of nanoparticles with higher magnification. Expecially for AgPt nanoparticles the roughness of surface should be commented. Is it due to metals or to the organic ligand (BPEI)?

3) A yield of the reaction should be introduced and commented. Is the reduction complete?

4) No comments are reported on the stability. I suggest the authors to introduce and comments the stability issues, introducing the measure of Z-potentials.

In the Supporting Informations file:

I suggest the author to introduce an index to help the reader. Please check: Fig S8 misses the average size

Author Response

Dear Dr. Nevena Chen

Editor of Materials

Dear Dr. Nevena Chen

Many thanks for your e-mail dated Dec. 31th, 2019. We have carefully considered the reviewer’s comments and revised our manuscript accordingly. The changes are highlighted with yellow pen in the revised manuscript. Here we would like to address his/her comments:

Review #2

The paper from Zengmin Tang et al., titled “Facile Aqueous-Phase Synthesis of Bimetallic (AgPt, AgPd, and CuPt) and Trimetallic (AgCuPt) Nanoparticles”, presented a water-based co-reduction methodology to afford bi and tri-metallic nanoparticles. The authors present an alternative route respect to the known protocols which exploit the use of L-ascorbic acid and BPEI to synthesize metallic nanoparticles in short reaction time and at not high temperature. The topic is particularly hot, considering the numbers of different application in which the nanoaprticles can be employed. Therefore, I suggest the publication of the presented paper in Materials after some revisions:

In the Materials and Methods part:

Comment 1. How the nanoparticles are purified? Please comment on it

Respose 1. We would like to thank this reviewer’s kind comment. The nanoparticle was purified by this process. After heating at the same temperature for 10 min, the resulting mixture was cooled down to room temperature and transferred into 50 mL tube, and centrifuged at 8000 rpm for 10 min to get products precipitate. The products precipitate was purified by repeat centrifugation and washing three times with water to remove excess stabilizer BPEI (Please refer line 81-85).

Comment 2. The size is measured by TEM? Please describe in details. How many nanoparticles are considered?

Response 2. We would like to thank this reviewer’s kind comment. We have added method to the revised manuscript (Please refer line 112 and 113).

Common 3. Describe the method used to prepare the sample for ICP.

Response 3. We would like to thank this reviewer’s kind comment. In the revised manuscript, the method used to prepare the sample for ICP have been added (Please refer to line 89-91)

Common 4. the size of the nanoparticles must be reported with their standard deviation, for any kind of nanoparticles.

Response 4. We would like to thank this reviewer’s kind comment. In this modified manuscript text, the average size of each samples have been present by the format of ab ± cd nm (please refer lines 123, 124, 153, 185 and 205).

Common 5. I suggest introducing at least one TEM image for kind of nanoparticles with higher magnification. Especially for AgPt nanoparticles the roughness of surface should be commented. Is it due to metals or to the organic ligand (BPEI)?

Response 5. We would like to thank this reviewer’s kind comment. In this revised manuscript, TEM images (Figure X2) of AgPt-6.7, Ag Pd-1, CuPt-10, and AgCuPt with high resolution have been added. (Please refer line 136-138 and Figure S4; line 157 and 158 and Figure S8; line 190 and 191 and Figure S11; line 214 and 215 and Figure S14).

Figure X2. TEM images of (a) and (b) AgPt-6.7, (c) and (d) Ag Pd-1, (e) and (f) CuPt-10, (g) and (h) AgCuPt at high resultion.

Comment 6. A yield of the reaction should be introduced and commented. Is the reduction complete?

Response 6. We would like to thank this reviewer’s kind comment. We have calculated the yield of purified nanoparticles and added into this revised manuscript (please refer line 129 and 130; line 160-162; line 194-196; line 214-216, and Table S1.).

Table X1 Yield of each component in each multi metallic nanoparticles

Sample

Yield (%)

Pt%

in NPS

Sample

Yield (%)

Pd%

in NPS

AgPt-1

Pt

23.8%

30.7%

AgPd-0.66

Pd

31.8%

34.3%

Ag

97.3%

Ag

97.8%

AgPt-2

Pt

29.2%

21.24%

AgPd-0.8

Pd

20.1

21.7%

Ag

96.8%

Ag

96.9%

AgPt-5

Pt

38.4%

12.85%

AgPd-1

Pd

21.3%

17.5%

Ag

96.9%

Ag

97.4%

AgPt-6.7

Pt

41.0%

10.12%

AgPd-1.3

Pd

2.9%

2.1 %

Ag

97.3%

Ag

95.9%

AgPt-10

Pt

30.7

5.58%

CuPt-20

Pt

83.1%

Ag

95.9%

Cu

34.0%

AgPt-40

Pt

26.7.7%

1.24%

CuPt-10

Pt

81.5%

Ag

96.89%

Cu

52.1%

AgCuPt

Pt

88.7%

CuPt-4

Pt

82.1%

Ag

96.4%

Cu

77.5%

Cu

58.7%

Comment 7. No comments are reported on the stability. I suggest the authors to introduce and comments the stability issues, introducing the measure of Z-potentials.

Response 7. We would like to thank this reviewer’s kind comment. We have introduced the Z-potentials to evaluate its stability in this revised manuscript. The Z-potential of each sample was shown in table X2 (please refer line 217-221, Table S2, and reference 29).

Table X2 zeta potential of each sample

Sample

Z-potential (mV)

Sample

Z-potential (mV)

AgPt-1

23.7

AgPd-0.66

25.9

AgPt-2

21.8

AgPd-0.8

28.3

AgPt-5

24.5

AgPd-1

27.2

AgPt-6.7

22.9

AgPd-1.3

26.7

AgPt-10

20.5

CuPt-4

28.1

AgPt-40

21.3

CuPt-10

27.4

AgCuPt

20.6

CuPt-20

29.6

Comment 8. In the Supporting Information file, I suggest the author to introduce an index to help the reader. Please check: Fig S8 misses the average size.

Response 8. We would like to thank this reviewer’s kind comment. To help the reader, Figure list in supporing informatin file was made, and avergae size has been added in Fig.S10 (please refer line 9-37, and line 105 (Figure S10) in supporting information file).

Thank you very much for many appropriate and valuable comments. I am sure that these comments improved significantly the quality of the manuscript.

Sincerely yours,

Taekyung Yu
